# The Altered Functions of Shelterin Components in ALT Cells

**DOI:** 10.3390/ijms242316830

**Published:** 2023-11-27

**Authors:** Yanduo Zhang, Kailong Hou, Jinkai Tong, Haonan Zhang, Mengjie Xiong, Jing Liu, Shuting Jia

**Affiliations:** Laboratory of Molecular Genetics of Aging and Tumor, Medical School, Kunming University of Science and Technology, 727 Jing Ming Nan Road, Kunming 650500, China; yanduozhang@163.com (Y.Z.); 20213136001@stu.kust.edu.cn (K.H.); 20212136004@stu.kust.edu.cn (J.T.); 20222136028@stu.kust.edu.cn (H.Z.); 20232134007@stu.kust.edu.cn (M.X.)

**Keywords:** telomere, shelterin, replication stress, DNA replication, DNA repair

## Abstract

Telomeres are nucleoprotein complexes that cap the ends of eukaryotic linear chromosomes. Telomeric DNA is bound by shelterin protein complex to prevent telomeric chromosome ends from being recognized as damaged sites for abnormal repair. To overcome the end replication problem, cancer cells mostly preserve their telomeres by reactivating telomerase, but a minority (10–15%) of cancer cells use a homologous recombination-based pathway called alternative lengthening of telomeres (ALT). Recent studies have found that shelterin components play an important role in the ALT mechanism. The binding of TRF1, TRF2, and RAP1 to telomeres attenuates ALT activation, while the maintenance of ALT telomere requires TRF1 and TRF2. POT1 and TPP1 can also influence the occurrence of ALT. The elucidation of how shelterin regulates the initiation of ALT remains elusive. This review presents a comprehensive overview of the current findings on the regulation of ALT by shelterin components, aiming to enhance the insight into the altered functions of shelterin components in ALT cells and to identify potential targets for the treatment of ALT tumor cells.

## 1. Introduction

Telomeres are composed of DNA repeat sequences TTAGGG and a protein complex called shelterin that binds stably to the ends of chromosomes. They protect the linear chromosome ends from being recognized as DNA damage, and prevent the chromosomes from activating DNA damage repair pathways and aberrant repair, thus maintaining genomic stability [1]. The shelterin complex consists of TRF1 (telomere repeat-binding factor 1), TRF2 (telomere repeat-binding factor 2), POT1 (protection of telomeres 1), TPP1 (telomere-binding protein POT1-interacting protein 1), TIN2 (TRF1-interacting nuclear protein 2), and RAP1 (repressor/activator protein 1). TRF1 and TRF2 specifically bind to telomeric double-stranded DNA, while POT1 binds to telomeric 3′-overhang single-stranded DNA and interacts with TPP1. TIN2 connects TRF1, TRF2, and POT1-TPP1. RAP1 is associated with telomeric DNA through its interaction with TRF2. Together, these shelterin components inhibit the activation of ataxia telangiectasia-mutated (ATM) kinase, ataxia telangiectasia and Rad3-related (ATR) kinase, homologous recombination (HR), and nonhomologous end joining (NHEJ) to maintain genomic homeostasis [2,3,4].

Human telomeres gradually shorten with each round of cell division until they reach a critical point that leads to the activation of DNA damage response (DDR) and cellular programs that prevent additional cell divisions. However, tumor cells can maintain their telomere length and keep their genome stable by activating telomere maintenance mechanisms (TMMs). The majority of tumor cells often reactivate telomerase, which consists of two components: a protein called TERT which acts as a reverse transcriptase, and an RNA component called TERC. By using TERC as a template, telomerase synthesizes and adds TTAGGG repeats to chromosome ends to preserve telomere length [5]. However, some immortalized cell lines and tumors maintain the telomere length in the absence of telomerase activity by a mechanism referred to as alternative lengthening of telomeres (ALT). The characteristics of ALT cells that have been explored to date include the appearance of heterogenous telomere length; high rates of telomeric sister chromatid exchange (T-SCE) [6]; the presence of extra-chromosomal telomeric DNA, such as C-Circle and T-Circle; and the presence of ALT-associated PML (promyelocytic leukemia) bodies (APBs) [7]. The onset of ALT is believed to depend on the escalation of replication stress, which eventually induces single-strand breaks (SSBs) or DNA double-strand breaks (DSBs), activating the ALT mechanism to repair DNA damage [8]. ALT requires a 3′-overhanging end and the homologous recombination proteins RAD51 or RAD52 to search for and invade a homologous strand [9,10,11]. The strand invasion leads to the formation of a recombination intermediate called a D-loop (DNA displacement loop), which is then disassembled by the BTR complex (containing BLM, TOP3α, RMI1, and RMI2) or the structure-specific endonuclease SMX (SLX1-SLX4, MUS81-EME1, and XPF-ERCC1) to allow DNA synthesis and telomere length maintenance [11,12,13,14,15].

The involvement of the shelterin complex in telomere protection and maintenance has been extensively explored in telomerase-positive tumor cells. However, the role of shelterin in ALT cells has not been deeply studied. Evidence has emerged suggesting an important role for shelterin alteration in the ALT mechanism. For instance, members of the shelterin complex have been shown to recruit proteins that participate in maintaining ALT telomeres. The shelterin components TRF1, TRF2, TIN2, and RAP1 are important for APB formation, and TRF1 and TRF2 are shown to interact with PML, suggesting that they may be involved in telomere extension in ALT cells [16,17] (Figure 1). Interactions of NBS1 with TRF1 or TRF2 have been found to be required for recruiting the DNA repair factors to APBs. The TRF1 and TRF2 interact protein TRIM28/KAP1 is preferentially located on the telomeres of ALT cells and regulates telomere maintenance in ALT cells [18]. In addition, post-translational modifications of TRF1 and TRF2, particularly SUMOylation, are found to occur at dysfunctional telomeres where they cause ALT [19]. However, it is also indicated that the partial depletion of TRF2 at telomeres is also associated with APB formation [20], which is probably due to ALT tumor cells requiring a certain level of DNA damage to initiate HR. It is also possible that the shelterin complex can provide a favorable environment for the emergence of ALT by inhibiting telomerase activity. It is shown that POT1a’s ability to outcompete POT1b for access to the 3′G-overhang inhibits telomerase recruitment to telomeres in mouse cells [21]. In addition, TRF2 interacts with the non-histone chromatin-associated protein HMGB1 and retains it in the nucleus, which inhibits its function as an activator of autophagy in the cytosol, possibly promoting cell growth in the initial phases of tumorigenesis [22]. And HMGB1 gene knockout results in reduced telomerase activity and telomere in mouse embryonic fibroblasts (MEFs) [23], so that it can be speculated that shelterin components can regulate telomerase activity through some unknown pathways, but this needs to be further explored. To sum up, shelterin is a versatile protein complex that safeguards telomeres from damage and repair, as well as modulates the telomere maintenance mechanisms in ALT cells. Further characterization of shelterin function in the ALT mechanism will contribute to the development of ALT-targeted therapeutic strategies. Here, we reviewed the most recent research advances to provide insights into the role of the shelterin complex involved in the ALT mechanism.

## 2. TRF1 and TRF2

The duplex telomere-binding proteins TRF1 and TRF2 play an important role in promoting the replication of telomeric DNA and inhibit inappropriate activation of DNA damage response at the telomere to maintain telomere homeostasis. Telomeres are late-replicating regions of chromosomes that are difficult to replicate and usually form complex secondary structures, such as the telomere loop (T-loop), RNA-DNA hybrids (R-loop), and the G-quadruplex (G4). These secondary structures hinder the synthesis of telomeric DNA by telomerase, while TRF1 and TRF2 can resolve these replication stresses and promote the smooth replication of telomeric DNA. It has been reported that the BUB3-BUB1 complex cooperates with TRF1 and TRF2 to recruit BLM helicase to unwind G4 and promote the replication of telomeric DNA [24]. In addition, TRF1 and TRF2 can also recruit the helicase RTEL1 to the telomeres during the S phase, thereby unwinding the T-loop and allowing the replication of telomeric DNA [25,26] (Figure 2a). TRF1 is further found to be PARylated by PARP1 (Poly ADP-ribose polymerase 1), which reduces its binding to telomeric duplex DNA and facilitates the progression of the replication fork. Subsequently, the PARP1/TRF1 complex remains associated with single-stranded DNA and recruits BLM and WRN helicase to resolve secondary structures formed on the G-rich lagging strand [27] (Figure 2b). In addition, TRF1 helps telomere replication by recruiting Timeless, a component of the replisome fork protection complex, to telomeres to prevent replication fork stalling [28]. TRF2′s promotion of telomere replication has also been reported. TRF2 can bind and facilitate the loading of the origin recognition complex (ORC) and the replicative helicase MCM complex on telomeric DNA to promote telomere replication [29] (Figure 2c).

Deletion of either TRF1 or TRF2 activates ATM-dependent classical nonhomologous end joining (C-NHEJ) repair, revealing an important function of TRF1/TRF2 evolved to protect telomeres from being recognized as damaged sites primarily by inhibiting the DDR pathway [1,30,31]. Some studies have demonstrated that TRF2 promotes telomeric 3′-overhang single-stranded DNA invasion into double-stranded DNA to form a noose-like T-loop, which inhibits MRN complex (Mre11/Rad50/Nbs1) binding to telomeric 3′-overhang ends and further inhibits an ATM response [32,33]. In addition, TRF2 directly blocks Ku70/Ku80 heterotetramerization to inhibit C-NHEJ at telomeres and prevent telomere end fusion, which suggests that TRF2 can also block C-NHEJ independently of the DDR pathway [34] (Figure 2d). TRF2 also regulates TERRA (telomere repeat-containing RNA, which is transcribed from telomeric DNA repeats) formation to modulate the level of replication stress at telomeres. It has been reported that TRF2 recruits TCOF1 (Treacle ribosome biogenesis factor 1) to telomeres during the S phase to inhibit telomere transcription by binding and repressing RNA PolII, thereby reducing TERRA expression and replication stress [35] (Figure 2e).

### 2.1. TRF1 and TRF2 Promote Telomere Maintenance in ALT Cells

ALT is thought to be an HR-dependent telomere maintenance mechanism; therefore, ALT tumor cells tend to have a high level of DNA damage repair activity. Due to the DDR inhibition function of TRF1 and TRF2, the dissociation or transient departure from telomeres of TRF1 and TRF2 is likely necessary for initiation or maintenance of ALT. However, more recent findings suggest that TRF2 is also required for ALT cell proliferation [36]. Knockdown of TRF2 in ALT cells resulted in cellular senescence, substantial telomere loss, and shortening [36]. Similarly, other studies have reported that TRF2 localization at telomeres maintains ALT cell activity by inhibiting C-NHEJ, which is conducive to telomere recombination in ALT cells [37,38]. The binding of TRF2 at the ALT telomere is regulated by MMS21 (sumoylation E3 ligase)-dependent SUMOylation. SUMOylated TRF2 binds stably to telomeres to inhibit C-NHEJ, thus avoiding telomere fusion. In this pattern, MMS21 is stabilized by BRCC3, which is recruited by PARP1. Therefore, PARP1 inhibitor (PARPi) disrupted telomere localization of TRF2 and activated C-NHEJ-mediated telomere fusion, resulting in cell death (Figure 3a), making it a promising strategy to target ALT-dependent tumors [38]. Chromosomal double-strand breaks in mammalian cells can be repaired by either HR or C-NHEJ, which are mutually exclusive pathways that compete for access to a DNA damage site [39]. The possible mechanism by which TRF2 may enhance ALT generation is to suppress C-NHEJ and favor HR activity [20]. It has also been reported that TRF2 is involved in regulating the heterochromatinization of telomeres, which is associated with ALT activity [18,40]. TRIM28 is recruited by TRF1 and TRF2 to protect telomere histone methyltransferase SETDB1 from degradation, thus maintaining the H3K9me3 heterochromatin state of ALT telomere DNA. Knockdown of TRIM28 in ALT cells reduces the level of C-Circle and delays cell growth [18]. However, in other studies, TRF2 appears to be mutually exclusive from telomere heterochromatinization. Telomeric TR4/COUP-TF2 recruits the NuRD-ZNF827 complex to both deplete shelterin and restore chromatin compaction by histone hypoacetylation [41]. Possibly, TRF2 binding may be a dynamic process and subsequently affects the DDR response and ALT activity [18,20,41]. One most recent study verified the depletion of TRF2 results in elevated levels of TERRA, which can be translated to generate dipeptide repeat proteins: valine–arginine (VR)_n_ and glycine–leucine (GL)_n_. VR is found to locate at the DNA replication fork and the Holliday junction. Meanwhile, a higher level of VR dipeptide protein is detected in the ALT line U2OS [42]. This finding provides a new perspective to explain the mechanism by which TRF2 regulates ALT by modulating replication stress. TRF2 regulates ALT telomere stability by alternating telomere shedding and re-enrichment, which modulates the DDR response, the recruitment of DNA damage repair factors, and the compaction of telomeres. This cycle blocks the ATM pathway and C-NHEJ pathway-mediated telomere fusion [20,43,44] (Figure 3b).

Similarly, TRF1 also plays an important role in ALT activation primarily by enhancing replication stress at telomeres [42]. Studies have reported that loss of TRF1 in mouse embryonic fibroblasts (MEFs) causes an increase in TERRA expression, which increases replication stress at telomeres and recruit proteins involved in DNA damage repair and recombination processes and contributes to ALT-associated phenotypes such as APBs and T-SCE [42]. However, a certain level of TRF1 is needed to maintain the ALT telomere [43]. Polη has been proven to be recruited to ALT telomeres through TRF1 to manage replicative stress at ALT telomeres. Polη inhibits activation of ATM on ALT telomeres, thereby preventing telomere shortening and senescence [20]. In addition, the phosphorylation of TRF1 by MRE11 and BRCA1 is required for the accumulation of APBs in the G2 phase [43] (Figure 3c). Taken together, the regulation of TRF1 and TRF2 is thought to be necessary for the initiation of ALT, mainly because ALT telomeres require both protection by shelterin and its modulated DDR initiation and DNA synthesis.

### 2.2. TRF1 and TRF2 Promote Proper Telomere Localization in APBs and Telomere Replication in ALT Cells

APBs are the main places where ALT cell telomeres undergo reorganization and lengthening. The MMS21 ubiquitin ligase within the SMC5/6 complex mediates the SUMOylation of TRF1 and TRF2, which facilitates telomere proper localization to PML and enhances ALT telomere recombination, basically through two possible mechanistic models [45]. In the recruitment model, MMS21 mediates the SUMOylation of TRF1 and TRF2 in the nucleoplasm, which leads to the subsequent localization of telomeres to PML bodies for telomere recombination and elongation. In the maintenance model, telomeres and shelterin first localize in APBs in an MMS21-independent manner, and then MMS21 SUMOylates TRF1 and TRF2 to maintain telomere recombination and elongation at APBs [45]. The exact molecular mechanism of this process remains elusive, but a synergistic interplay of both mechanisms in ALT cells may create a positive feedback loop that drives ALT activation [45] (Figure 4a). Similarly, SUMOylation of TRF2 by PIAS4 is reported to increase the recruitment of DDR-related proteins to APBs and promote break-induced replication (BIR) [46] (Figure 4b). In the telomere recombination process, ALT cells require efficient cleavage of recombination intermediates to preserve telomere length. TRF2 modulates the cell-cycle-specific activity of MUS81 endonuclease to cleave recombinogenic intermediates in ALT cells [47,48] (Figure 4c).

As a conclusion, TRF1 and TRF2 have dual roles in ALT regulation. On the one hand, their binding protects telomere integrity and maintains DNA replication by recruiting ALT-related proteins. In addition, their transient dissociation from telomere may also be necessary to trigger DDR and initiate ALT, and this process is precisely regulated by the cell cycle, SUMOylation, etc.

## 3. RAP1

RAP1, a telomere-binding protein with a crucial role in telomere maintenance, enhances the specificity and stability of TRF2–telomere interactions through the Rap1 C-terminal domain (RCT) TRF2-binding domain [49]. In mouse cells, RAP1 absence does not compromise the recruitment of other telomere-binding proteins to telomeres or elicit DDR, implying that TRF2 suffices to repress the ATM kinase and C-NHEJ pathways at telomeres. However, the deletion of RAP1 increases the level of T-SCE in cells, suggesting that RAP1 plays a key role in inhibiting telomere homology-directed repair [50].

### 3.1. Depletion of RAP1 Accumulates Replication Stress at Telomeres and Triggers ALT

It is reported that spontaneous DNA damage at telomeres may provide the substrate for BIR-mediated ALT telomere elongation, and TERRA transcription is believed to be a major trigger of replication-stress-associated telomere instability and, in turn, BIR-mediated telomere elongation in ALT cells [51]. RAP1 has been identified to repress the transcription of TERRA, alleviate DNA replication stress and DNA damage at telomeres, and thwart the onset of ALT [52,53,54]. Previous reports have shown that yeast-lacking telomerase activity exhibits progressive telomere shortening. Conversely, when RAP1 is evicted from telomeres, it elicits upregulation of TERRA, which extends telomere length in yeast and augments yeast proliferation [52]. Likewise, in ALT cells, ablation of the telomere-specific RNaseH1 nuclease triggers TERRA accumulation, leading to increased replication stress. Overexpression of RNaseH1 reduces R-loop levels and impairs the capacity of ALT telomeres for recombination, resulting in telomere attrition [55]. This suggests that RAP1 can modulate the level of replicative stress and thus regulate ALT activity by controlling TERRA expression.

It has also been demonstrated that RAP1 or XRCC1 downregulated by mutant IDH1 contributes to the ALT phenotype in ATRX loss glioma. RAP1 downregulation caused telomeric dysfunction, which ensured persistent endogenous telomeric DNA damage. In addition, XRCC1 silencing suppressed alternative nonhomologous end joining (A-NHEJ) and permitted ATRX-deficient cells to use HR and ALT to process dysfunction telomeres and escape cell death [54].

### 3.2. RAP1 Coordinates the Spatial Localization of Telomeres and the Epigenetic State of Telomere Chromatin to Prevent the Generation of ALT

Previous studies have demonstrated that chromatin employs distinct pathways to cope with DNA damage and that severe lesions trigger the relocation of chromatin to the perinuclear region for repair [56,57]. Telomere repair is also associated with nuclear localization of chromatin in ALT cells. RAP1 interacts with the N-terminal domain of SUN1, a crucial component of the mammalian LINC complex that consists of trans-membrane SUN-domain proteins, through its helical domain to tether telomeres to the nuclear envelope, by which RAP1 reduces the capacity of telomeric DNA to undergo homologous recombination in the nucleus. However, disrupting the interaction of RAP1–SUN1 does not interfere with APB formation, which suggests the existence of another SUN1-dependent telomere anchorage pathway [58] (Figure 5a). It has also been reported that a 53BP1-LINC complex-dependent homologous sequence search is performed in the nucleus of telomeric DNA during the G2 phase of the cell cycle, which enhances the homologous recombination of telomeric DNA [59]. Hence, the tethering of telomeres by RAP1–SUN1 at the nuclear envelope not only curtails the capacity of telomeres to search homologous strands but also attenuates the level of DNA damage and thereby precludes the 53BP1-facilitated homologous recombination of telomere DNA. This mechanism may be the most plausible explanation for how RAP1 modulates ALT [58,59].

In addition to “anchoring” telomeres to the nuclear membrane to inhibit ALT, RAP1 has been reported to maintain the expression of ATRX and DAXX to stabilize chromatin to inhibit ALT [60]. In the nucleus, RAP1 is SUMOylated by PIAS1 (an E3 ubiquitin ligase recruited to the SLX4 complex and activated by SLX4IP) and dissociates from TRF2. SUMOylated RAP1 crosses the nuclear pore to form a complex with the β-subunit of NF-κB kinase (IKKβ) in the cytoplasm, followed by the activation of the IKK and NF-κB signaling pathways. NF-κB inhibits TERT expression while inducing Jagged-1 expression and activating the Notch signaling pathway in neighboring cells, which induces epigenetic silencing of ATRX and DAXX, modifying the chromatin state and cooperating with low TRET expression to enhance ALT activity [60]. Suppressed Notch signaling not only reduces the levels of APBs and blocked C-Circle formation in ALT cells, but also hinders cell proliferation. Thus, RAP1 and SLX4IP can cooperatively regulate ALT development, while NF-κB and Notch can also serve as novel targets in ALT cancer therapy [60] (Figure 5b).

## 4. Discussion

Accumulating evidence indicates that the shelterin complex plays an important role in the regulation of replication stress, DNA repair and replication, and the DNA damage response in ALT cells. In addition to TRF1, TRF2, and RAP1, other shelterin components have also been reported in recent years to be associated with the regulation of the ALT mechanism. The 3′-overhang of telomere is normally coated by TPP1/POT1. It is possible that TPP1/POT1 is relatively lacking in ALT telomeres to provide a classic single-stranded substrate for RAD51-associated homologous recombination [61]. How this TPP1/POT1-coated overhang may be converted into a recombination filament is unknown. However, POT1 is reported to bind to SUN1 and anchor telomeres to the nuclear membrane, inhibiting HR in *Caenorhabditis elegans* telomere [62]. Depleting telomerase and POT1/POT2 upregulated TERRA expression and activated the ALT mechanism in *C. elegans* [63], suggesting that POT1 and POT2 repress ALT. In addition, TPP1^△OBRD^, an OB-RD structural domain mutation in TPP1, impairs telomerase activity at telomeres and induces elevated levels of ALT phenotypes (such as APBs and C-Circle) [64]. However, studies have shown that knocking down TPP1 in ALT cells causes cell apoptosis and telomere shortening [65]. Furthermore, in differentiated mouse embryonic stem cells (mESCs), TIN2 mutation impairs ATRX and DAXX localization at PML, which augments replicative stress in cells and elicits an ALT-related phenotype [66]. It may therefore be that ALT requires an exact series of events that includes a generalized induction of telomere dysfunction coupled with alterations in DNA repair pathways that favor HR.

Another possible point that needs to be noticed is that shelterin components may work synergistically to regulate ALT. The most recent study demonstrated that loss of PPM1D activity leads to hyper-phosphorylation of TRF2 and promoted recruitment of TIN2 and TPP1 to the telomeres, which prevents recruitment of 53BP1 to the telomeric DSBs [67], possibly increasing chances of ALT occurrence. In addition, recent reports demonstrated that some variant telomere repeats, such as A/GGGTCA, have been shown to be enriched in telomeres of ALT cells. These non-classical telomere repeats provide the binding sites for some orphan nuclear receptors (NRs), such as COUP-TFII and TR4, which may compete with the shelterin complex in binding to telomere, therefore disrupting the inhibition of the DDR in ALT telomeres [41]. In addition, it is also proved that COUP-TFII/TR4 recruits FANCD2 to ALT telomeres and induces the DNA damage response by recruiting endonuclease MUS81 and promoting the loading of the PCNA-POLD3 replication complex for telomere maintenance [68]. Experimental demonstration of the dynamic interaction between shelterin components and telomeres may need to further explain the facilitating of intertelomeric recombination in ALT-positive cells.

Although the exact mechanism of shelterin components in regulating ALT occurrence is still elusive, it is promising that we have verified that altering shelterin at the telomere can indeed affect ALT. In conclusion, a comprehensive analysis of shelterin components in ALT cells is needed to further understand the role of shelterin in telomere HR and recruitment to PML bodies and to expand new targets for the treatment of ALT tumor cells.

## Figures and Tables

**Figure 1 ijms-24-16830-f001:**
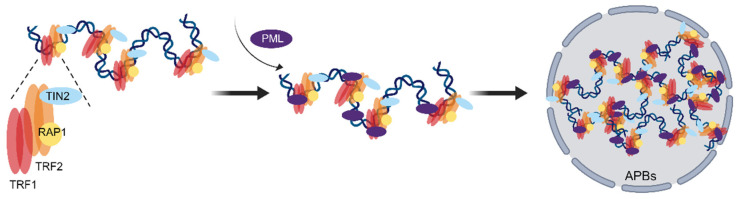
Shelterin is involved in the formation of ALT-associated promyelocytic leukemia bodies (APBs). Where red represents TRF1, orange represents TRF2, yellow represents RAP1, blue represents TIN2, and purple represents PML.

**Figure 2 ijms-24-16830-f002:**
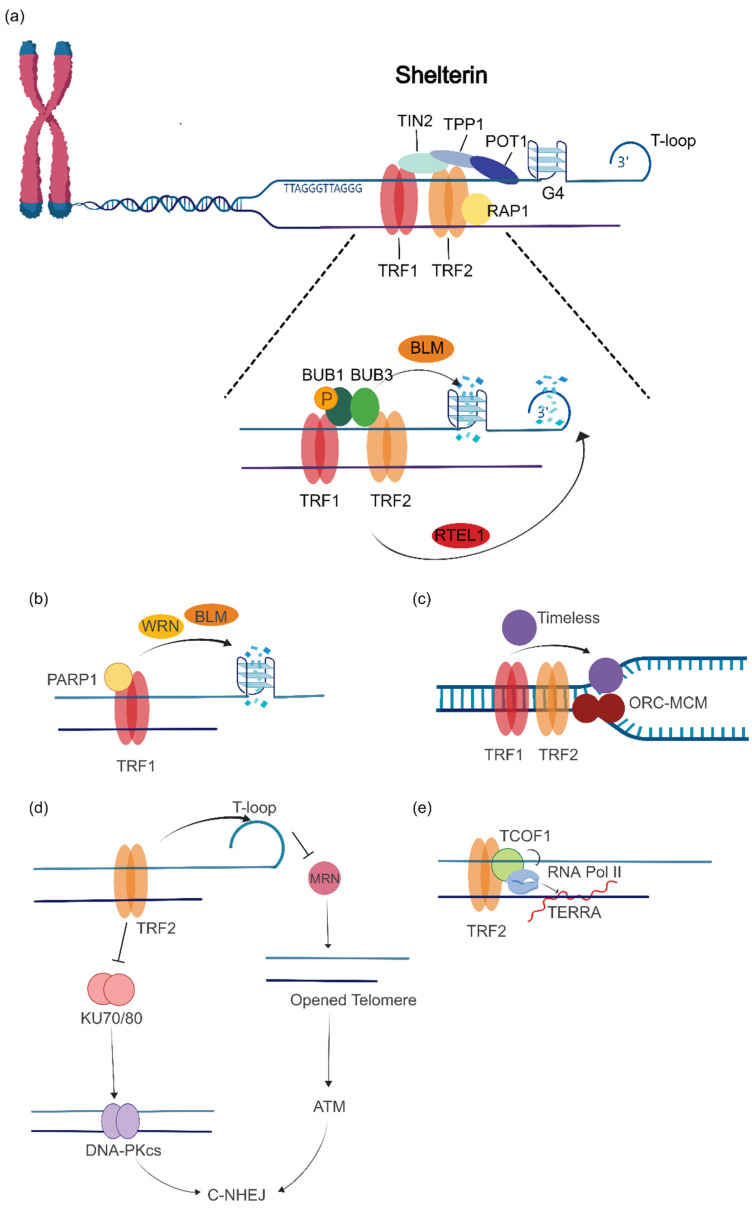
TRF1 and TRF2 resolve replication stress and promote telomeric DNA replication. (**a**) TRF1 and TRF2 promote phosphorylation of BUB1, which recruits BLM to resolve G-quadruplex (G4) and RTEL1 to resolve the telomere loop (T-loop) to promote telomeric DNA replication. (**b**) TRF1 is PARylated by PARP1, and subsequently, the PARP1/TRF1 complex recruits BLM and WRN helicases to resolve G4 to promote telomeric DNA replication. (**c**) TRF1 recruits Timeless to telomeres, preventing replication fork stalling and facilitating telomere replication. (**d**) TRF2 inhibits the activation of ataxia telangiectasia-mutated (ATM) kinase and classical nonhomologous end joining(C-NHEJ) by sequestering the telomere end in the T-loop, thereby preventing the end-loading of the Mre11/Rad50/Nbs1 (MRN) complex and the Ku70/80 heterodimer. (**e**) TRF2 recruits TCOF1 to telomeres during the S phase to repress telomeric transcription and reduce replication stress by binding and inhibiting RNA Pol II.

**Figure 3 ijms-24-16830-f003:**
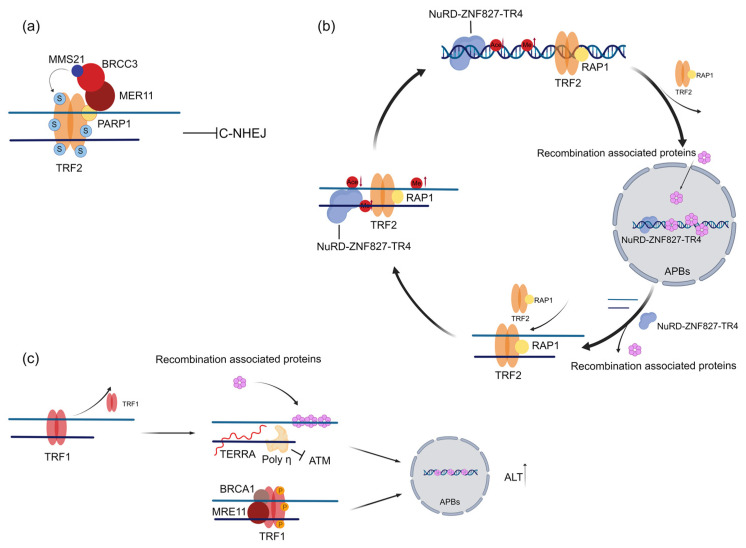
TRF1 and TRF2 promote telomere maintenance in ALT cells. (**a**) SUMOylated TRF2 binds ALT telomeres and blocks telomere fusion by the MMS21-BRCC3-MRE11-PARP1 complex. (**b**) The telomere chromatin remodeling is regulated by the TR4-NURD-ZNF827 complex, which contributes to the dynamic binding of shelterin binding, homologous recombination (HR), and ALT activity. (**c**) TRF1 regulates telomeric transcription and replication stress by detaching from telomeres and recruiting Polη. The phosphorylation of TRF1 by MRE11 and BRCA1 is required for the accumulation of APBs.

**Figure 4 ijms-24-16830-f004:**
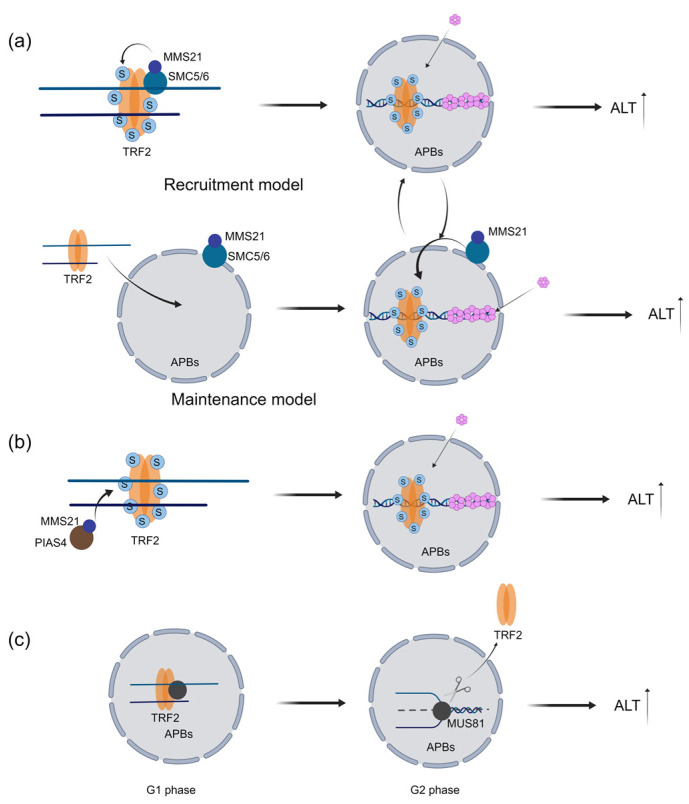
TRF1 and TRF2 promote proper telomere localization in APBs and telomere replication in ALT cells. (**a**) Two proposed models for how shelterin SUMOylation by MMS21 helps telomeres stay in APBs. In the recruitment model, MMS21 induces the SUMOylation of the shelterin complex within the nucleoplasm, leading to the subsequent attraction of telomeres to PML bodies for recombination and elongation. In the maintenance model, MMS21 is present in APBs independently of telomeres. When telomeres are brought to APBs, shelterin is SUMOylated by MMS21, which facilitates telomere recombination. (**b**) The SUMOylation of TRF2 mediated by PIAS4 and MMS21 enhances the recruitment of DDR-related proteins to APBs and facilitates break-induced replication (BIR). (**c**) In G1 phase, TRF2 binds to MUS81 and suppresses its nuclease activity within APBs. In G2 phase, TRF2 releases MUS81 in the APBs which activates MUS81 to cleave the recombination intermediates.

**Figure 5 ijms-24-16830-f005:**
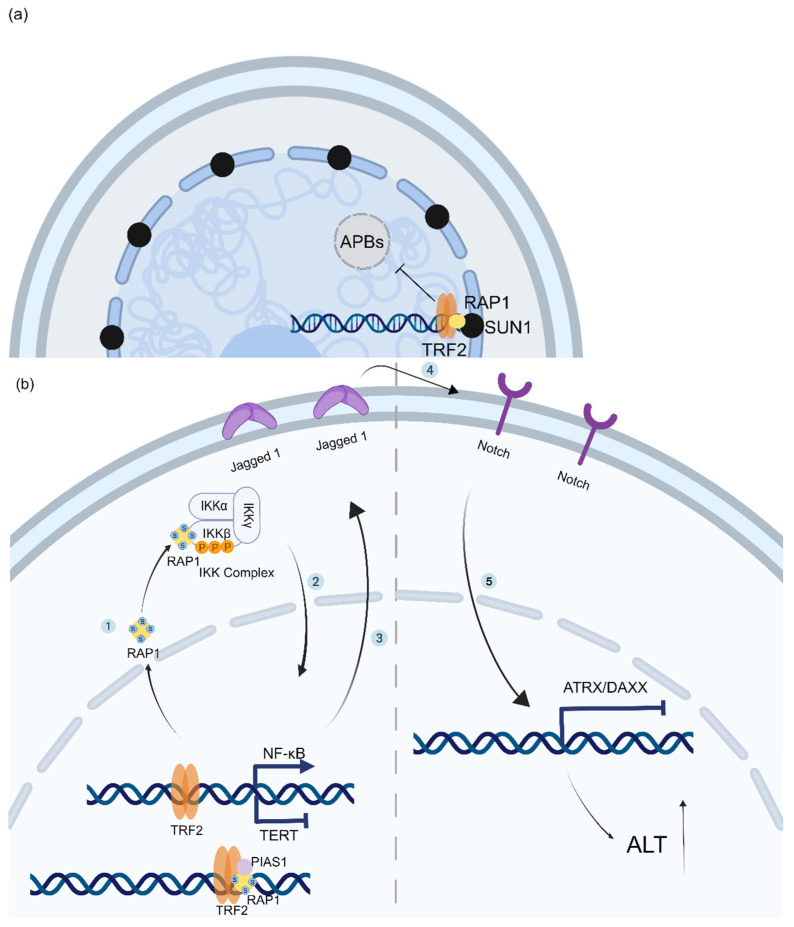
RAP1 coordinates the spatial localization of telomeres and the epigenetic state of telomere chromatin to prevent the generation of ALT. (**a**) The interaction between RAP1 and SUN1 contributes to telomere anchorage to the nuclear envelope, which inhibits the formation of ALT-associated PML bodies in ALT cells. (**b**) 1: In the nucleus of ALT cells, RAP1 is SUMOylated by PIAS1 and detaches from TRF2. 2 to 3: In the cytoplasm, SUMOylated RAP1 engages and phosphorylates the β-subunit of NF-κB kinase (IKK-β), thereby activating the NF-κB pathway, which modulates Jagged-1 transcription and TERT repression. 4 to 5: Jagged-1 binds to Notch receptors on neighboring cells, leading to epigenetic silencing of ATRX and DAXX.

## Data Availability

Not applicable.

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
