# Peer review of "The Altered Functions of Shelterin Components in ALT Cells"

_ijms, 2023, doi:10.3390/ijms242316830_

Round 1

Reviewer 1 Report

Comments and Suggestions for Authors

The authors of the manuscript titled: " Altered functions...............", Y. Zhang et al have given a comprehensive and well-written review of the Shelterin functions in the process of the alternative lengthening of the telomeres with original graphs and drawings. Before the paper is complete and ready for publication, I would advise the authors to proceed with certain improvements to the manuscript. 

The keywords should not overlap with, but enhance the title with suitable terms.

In lines 96-97, it would be clearer to state that:  Telomeres are late replicating regions of the genome in which a number of replicative mechanisms reside...

Line 142: HR stands for Homologous Recombination and should be explained where it appears first.

Line 182: explain what MEFs stand for, also in line 186 use a capital T at the beginning of the sentence.

In lines 200-204 and 300-308, there is no reference included.

In lines 208, 231, 235, and 290 explain what BIR, RCT, T-SCE, and IKK stand for at the point they appear first.

The sentence between lines 317-319 should be better rephrased to convey a more comprehensible message.

Comments on the Quality of English Language

The English language is comprehensible and readable but minor changes and definitions would improve the final appearance of the review. 

Author Response

Response to Reviewer Comments

  1. The keywords should not overlap with, but enhance the title with suitable terms.

         Response: Thanks for the reminder, we've changed the keyword so that it doesn't overlap with the title.

  1. In lines 96-97, it would be clearer to state that: Telomeres are late replicating regions of the genome in which a number of replicative mechanisms reside...

         Response: This section has been re-edited and highlighted in red on lines 98-101.

         Telomeres are late replicating regions of the chromosomes that are difficult to replicate and usually form complex secondary structures, such as the telomere loop (T-loop), RNA-DNA hybrids (R-loop), and the G-quadruplex (G4).

  1. Line 142: HR stands for Homologous Recombination and should be explained where it appears first.

         Response: Thanks, we explained the meaning of HR in lines 36-37 and marked it by red.

  1. Line 182: explain what MEFs stand for, also in line 186 use a capital T at the beginning of the sentence.

         Response: MEFs (mouse embryonic fibroblasts) has been explained in line 194, and marked it by red. And the “T” in line 186 has been removed.

  1. In lines 200-204 and 300-308, there is no reference included.

         Response: Thanks for the reminder, we've inserted the cited references and marked it by red.

  1. In lines 208, 231, 235, and 290 explain what BIR, RCT, T-SCE, and IKK stand for at the point they appear first.

         Response: Thanks to your reminder, we have made the appropriate changes in lines 220,244 and 316 respectively and marked them in red.

  1. The sentence between lines 317-319 should be better rephrased to convey a more comprehensible message.

         Response: Thanks, we have removed the sentence after much deliberation. Considering that to make it clear it would need to be expanded by a certain number of paragraphs in order to avoid more errors it has been removed.

Reviewer 2 Report

Comments and Suggestions for Authors

The work submitted for review by Yanduo Zhang et al. titled “The altered functions of shelterin components in ALT cells” is well written and potentially interesting. It touches on many phenomena and threats regarding telomere sequence especially ALT. The sources of information cited are new, however in line 147 the Authors raise an important aspect of induction of telomere dysfunction by knockdown TRF2, which is the cause of cell aging, shortening and loss of telomeres. The brand new researches by Taghreed M. Al-Turkia and Jack D. Griffith in a 2023 reports on the mechanism of formation of dipeptide proteins (VR, GL), which are the result of knockdown TRF2 i transkrypcji TERRA. This contributes to instability and inflammatory reaction in the body in addition to carcinogenesis. In lines 51-55 the Authors write that replication stress contributes to the generation of double-strand breaks (DSBs) at telomeres. Replication stress also causes  single-strand breaks (SSBs) in these sequences, which in turn causes telomere shortening.

Please complete the relevant information in the manuscript.

Author Response

Response to Reviewer Comments

  1. The work submitted for review by Yanduo Zhang et al. titled “The altered functions of shelterin components in ALT cells” is well written and potentially interesting. It touches on many phenomena and threats regarding telomere sequence especially ALT. The sources of information cited are new, however in line 147 the Authors raise an important aspect of induction of telomere dysfunction by knockdown TRF2, which is the cause of cell aging, shortening and loss of telomeres. The brand new researches by Taghreed M. Al-Turkia and Jack D. Griffith in a 2023 reports on the mechanism of formation of dipeptide proteins (VR, GL), which are the result of knockdown TRF2 transkrypcji TERRA. This contributes to instability and inflammatory reaction in the body in addition to carcinogenesis. In lines 51-55 the Authors write that replication stress contributes to the generation of double-strand breaks (DSBs) at telomeres. Replication stress also causes single-strand breaks (SSBs) in these sequences, which in turn causes telomere shortening.

         Response: Thanks for the reminder, we have made the appropriate revisions to the "In lines 51-55 the Authors write that replication stress contributes to the generation of double-strand breaks (DSBs) at telomeres. Replication stress also causes single-strand breaks (SSBs) in these sequences, which in turn causes telomere shortening" paragraph you mentioned in lines 52-54, and marked it by red

        The onset of ALT is believed to depend on the escalation of replication stress, which eventually induces single-strand breaks (SSBs) or DNA double-strand breaks (DSBs), activating the ALT mechanism to repair DNA damage.

        Response: Thanks for the reminder, we have made the appropriate revisions to the "The brand new researches by Taghreed M. Al-Turkia and Jack D. Griffith in a 2023 reports on the mechanism of formation of dipeptide proteins (VR, GL), which are the result of knockdown TRF2 transkrypcji TERRA. This contributes to instability and inflammatory reaction in the body in addition to carcinogenesis" paragraph you mentioned in lines 329-335, and marked it by red.

        Recent studies have shown that knockdown of TRF2 results in elevated levels of TERRA, and in ALT cells, the level of the highly charged repeated valine-arginine (VR)n amino acid sequence that produces TERRA also increases. VR is located at the DNA replication fork and the Holliday junction, and in addition to its carcinogenic effect, it also causes instability and inflammation in the body. This may lead to an environment conducive to ALT development.

Round 2

Reviewer 1 Report

Comments and Suggestions for Authors

Amendments and corrections have been fulfiled